# Drying Microalgae Using an Industrial Solar Dryer: A Biomass Quality Assessment

**DOI:** 10.3390/foods11131873

**Published:** 2022-06-24

**Authors:** Benjamin Schmid, Sofia Navalho, Peter S. C. Schulze, Simon Van De Walle, Geert Van Royen, Lisa M. Schüler, Inês B. Maia, Carolina R. V. Bastos, Marie-Christin Baune, Edwin Januschewski, Ana Coelho, Hugo Pereira, João Varela, João Navalho, Alexandre Miguel Cavaco Rodrigues

**Affiliations:** 1Necton S.A., Belamandil s/n., 8700-152 Olhao, Portugal; benjamin.schmid@necton.pt (B.S.); ibmaia@ualg.pt (I.B.M.); carolinavbastos@hotmail.com (C.R.V.B.); ana.coelho@necton.pt (A.C.); jnavalho@necton.pt (J.N.); 2Centre of Marine Sciences, Faculty of Sciences and Technology, Campus of Gambelas, University of Algarve, 8005-139 Faro, Portugal; jvarela@ualg.pt; 3GreenCoLab—Associação Oceano Verde, Campus of Gambelas, University of Algarve, 8005-139 Faro, Portugal; sofianavalho@greencolab.com (S.N.); peterschulze@greencolab.com (P.S.C.S.); lisaschueler@greencolab.com (L.M.S.); hugopereira@greencolab.com (H.P.); 4ILVO Flanders Research Institute for Agriculture, Fisheries and Food, Technology & Food Science, Brusselsesteenweg 370, 9090 Melle, Belgium; simon.vandewalle@ilvo.vlaanderen.be (S.V.D.W.); geert.vanroyen@ilvo.vlaanderen.be (G.V.R.); 5German Institute of Food Technologies (DIL e.V.), Prof.-von-Klitzing Str. 7, 49610 Quakenbrück, Germany; m.baune@dil-ev.de (M.-C.B.); e.januschewski@dil-ev.de (E.J.)

**Keywords:** microalgae, solar drying, freeze drying, proteins, pigments, fatty acids

## Abstract

Microalgae are considered a promising resource of proteins, lipids, carbohydrates, and other functional biomolecules for food and feed markets. Competitive drying solutions are required to meet future demands for high-quality algal biomass while ensuring proper preservation at reduced costs. Since often used drying methods, such as freeze or spray drying, are energy and time consuming, more sustainable processes remain to be developed. This study tested an indirect and hybrid solar dryer as an alternative to conventional freeze drying of industrially produced *Tetraselmis chui* and *Nannochloropsis oceanica* wet paste. The effects of the drying method on biomass quality parameters, including biochemical profiles, functional properties, and microbial safety, were assessed. No significant differences were found between the applied drying technologies for total proteins, carbohydrates, lipids, and fatty acid profiles. On the other hand, some pigments showed significant differences, displaying up to 44.5% higher contents in freeze-dried samples. Minor differences were also registered in the mineral profiles (<10%). Analyses of microbial safety and functional properties of the solar-dried biomass appear adequate for food and feed products. In conclusion, industrial solar drying is a sustainable technology with a high potential to preserve high-quality microalgal biomass for various markets at expected lower costs.

## 1. Introduction

The global food demand will increase by up to 70% due to the growing human population, reaching over 9.7 billion by 2050 [1]. Exploiting conventional food and feed sources intensifies critical environmental issues such as land degradation, deforestation, water pollution, greenhouse gas emissions, and water consumption [2,3,4]. Hence, finding sustainable and circular alternatives for global food industries is of utmost importance [5,6]. Large-scale productions of microalgae appear as a promising solution since they can be sited on non-arable land and do not necessarily require potable water sources, as seawater or wastewaters can be used [7,8,9]. Moreover, microalgae have a wide tolerance to environmental conditions and a relatively low carbon footprint, which can be further reduced using energy-efficient solutions and flue gases rich in CO_2_ [10,11].

Microalgae gained popularity as functional and “superfoods” due to their content in high-value biomolecules such as polyunsaturated fatty acids (PUFAs), pigments, and vitamins, among others [12,13]. These biomolecules are also valued for livestock, poultry, or aquaculture feed ingredients, making microalgae one of the most promising bioresources for sustainable mass production of foods and feeds [14,15]. For example, eukaryotic microalgal species such as *Chlorella vulgaris*, *Tetraselmis chui*, or *Nannochloropsis oceanica* are known for their potential as sustainable and innovative food and/or feed ingredients [16,17,18]. Moreover, photosynthetic prokaryotes such as the cyanobacterium Spirulina (*Arthrospira platensis*) are widely used in food- and feed-related industries [19]. Although the industrial production of these species is still developing, areal biomass productivities have already exceeded those of commercial terrestrial plants such as soybean by 2 to 20 times [20].

Within the production process of microalgal biomass, drying is considered one of the most cost-intensive and energy-requiring steps [21,22]. Freeze drying (lyophilization) is a commonly used technique to preserve biomass quality [23,24] but requires a high energy demand [23,25]. Furthermore, apart from the considerable acquisition costs, freeze drying involves energy-demanding steps, including low temperatures and pressure [26]. Similarly, spray drying also involves high energetic costs, and its use of high temperatures is known to affect biomass quality [27,28]. Therefore, alternative drying technologies are required to lower the current costs of microalgal biomass while ensuring high quality [29]. 

Solar drying is considered a more sustainable and cost-effective method [30,31], although direct biomass exposure to sunlight usually leads to the loss of functional properties [22,30,32]. Therefore, indirect solar-drying methods were developed, pairing the sustainable aspects of using the sun indirectly (heated air) while maintaining a good biomass quality [30,32]. Further process stability can be obtained by applying an indirect and hybrid solar dryer that uses sun irradiance as the primary heat source, coupled with a fan heater, a dehumidifier, and a ventilation system. If the external conditions are not favourable to reach the defined set points, the hybrid drying system consumes energy (dehumidifier and heater) to ensure optimum values. This setup provides constant temperatures and humidity inside the solar drier with low energy consumption. However, to our knowledge, no studies have been performed to assess the performance of these particular systems to dry microalgal biomass. 

This study compared the biochemical compositions and physical properties of solar- and freeze-dried *T. chui* and *N. oceanica* biomass. Moreover, microbial food safety analyses of the dried biomass were performed to evaluate the suitability for food products. Finally, functional properties (water- and oil-holding properties, foaming properties, emulsifying properties, solubility) were determined to assess food applicability further since these functionalities describe how an ingredient can provide and maintain food structure [33]. 

## 2. Materials and Methods

### 2.1. Microalgal Biomass Cultivation and Harvesting

*Tetraselmis chui* and *Nannochloropsis oceanica* biomass was obtained from Necton S.A. and grown in industrial tubular photobioreactors as part of a regular commercial production. Afterwards, wet biomass (paste) was packed in 1 kg plastic bags (310 × 220 × 30 mm), shaped into a flat plate, and stored at −20 °C. 

### 2.2. Drying Procedures and Moisture Analyses

The same production batches of *T. chui* and *N. oceanica* were used for solar and freeze drying. After drying, samples were milled with a commercial grinder (Drogheria MCD4, Eureka, Italy) to obtain a particle size < 300 µm. Afterwards, the biomass was packed and stored under a vacuum until further analyses.

#### 2.2.1. Freeze-Drying Trials

Freeze drying was used as the control (FDc) for both *T. chui* and *N. oceanica* biomass. Frozen plates of *T. chui* and *N. oceanica* (1 kg each) were freeze dried in a single drying procedure, according to the internal protocol of Necton S.A., in an industrial freeze dryer (F-50, FrozenInTime Ltd., York, UK). The frozen microalgal plates were loaded into the freeze dryer, which reached −18 °C after 38 min. Then, the pressure was decreased to 0.9 mbar and the temperature further decreased to −27 °C within 12 min. In the next step, the plates were dehydrated for 39 h with a gradual transition between −27 and 28.5 °C, keeping the vacuum at 0.9 mbar. The drying ended after five additional hours at 28.5–28.9 °C. In the last step, the plates were cooled to 4.8 °C and maintained in a vacuum for ~15 h. The total time of the freeze-drying procedure was approximately 60 h with a maximum load of 63 kg microalgal biomass.

#### 2.2.2. Solar Drying

Solar drying (SD) was performed in three separate trials (SD I, SD II, and SD III), starting on 25, 29, and 31 March 2021, lasting approximately 26 h each. Weather conditions (temperature, humidity, and radiation) were measured with Necton’s weather station (Watchdog Weather Station WD-2700, Spectrum Technologies Inc., Aurora, IL, USA) in 5 min intervals. Photosynthetically active radiation (PAR, in µmol/m^2^/s), provided by the weather station, was converted to solar radiation (*R_s_*, in W/m^2^) by dividing by the conversion factor of 2.02 [34]. Radiation averages were calculated from the values obtained during the trials, including the night period.

##### Hybrid Solar Dryer Characteristics and Operation

The solar dryer and its software were designed and built by BBKW Drying Solutions (Portugal). Ambient air was captured by two collectors (roof and wall collector) attached to the black-painted roof and the southwards-oriented sidewall of the dryer (Figure 1a,b). With solar irradiance, the air was warmed up. In-line duct fans blew the hot air from the collectors into the drying chamber while six other fans (TVM 24 D; 124 W, Trotec GmbH, Heinsberg, Germany) circulated the air inside the drying chamber. The drying operation was controlled by an algorithm that ensured the most effective drying process based on various temperature and humidity sensors (indoor and outdoor). This algorithm regulated air intake and discharge and the activation or deactivation of a dehumidifier (TTK 350 S; 70 L/ 24 h, 1.3 kW, Trotec GmbH, Heinsberg, Germany) and fan heater (TDS 20; 3.3 kW, Trotec GmbH, Heinsberg, Germany). After defining the set points for minimum humidity (25%, value above which the dehumidifier would start running) and maximum temperature (40 °C, value above which the heater would stop running), the system worked autonomously until the end of the drying process. BBKW specialised software provided indoor data (humidity and temperature) and the activity log for the dehumidifier and fan heater. 

##### Experimental Setup

For each of the three solar-drying trials, frozen plates of *T. chui* (3 × 1 kg) and *N. oceanica* (3 × 1 kg) were dried. Therefore, wet biomass of 9 kg was used per species in the three trials. The plates were placed inside an industrial hybrid solar dryer (BlackBlock, BBKW Solutions, Portugal) on top of plastic nets (1 mm^2^ mesh; Figure 1d). The nets were attached with paper clips to stainless-steel grids (9 mm^2^ mesh), elevated 8 cm above stainless-steel trays (94 × 34 × 2.4 cm). This structure ensured maximum airflow above and below the microalgal plates for efficient drying. In addition, foldable nets facilitated the final collection of the dried biomass. Biomass plates were placed in position 1 (Figure 1c) on shelves 1, 3, and 5 (Figure 1a) and were swapped during the drying process.

#### 2.2.3. Moisture Analyses 

The moisture loss was measured during the solar-drying experiments by weighing the trays with the biomass plates at different time points until no mass change was detected (BM 100M, Marques, Portugal). The algal biomass’s initial and final moisture content was then determined by drying the samples at 60 °C in a drying chamber (model no. 52301-45, Cole-Parmer Instrument Co., Chicago, IL, USA) until complete drying (approximately 36 h). Finally, the moisture contents (%) at a given time point during solar drying were calculated by fitting the weight loss to the initial and final moisture values. 

### 2.3. Biochemical Analyses and Physical Properties

Proximate composition, pigment, and fatty acid profiles were carried out at GreenCoLab (Associação Oceano Verde, Faro, Portugal). The German Institute of Food Technologies (DIL e.V.; Quakenbrück, Germany) analyzed mineral element contents, while the Flanders Research Institute for Agriculture, Fisheries and Food (ILVO; Melle, Belgium) determined functional properties and microbial safety.

#### 2.3.1. Proximate Composition

##### Protein Content

Protein content was determined by elemental analysis through the measurement of total nitrogen. Approximately 1 mg of dried biomass was weighed, and an element analyzer (Vario EL III, Elementar Analysensysteme GmbH, Langenselbold, Germany) was used to analyze C, H, and N according to the manufacturer’s methodology. Total nitrogen contents of solar- and freeze-dried *T. chui* and *N. oceanica* biomass were multiplied by the standard conversion factor of 6.25 to determine total protein contents [35,36,37].

##### Lipid Content 

Lipid contents were determined according to a modified Bligh and Dyer method [38], described in Pereira et al. [39]. First, dried microalgal biomass (10–20 mg) was homogenized with a mixture of methanol, chloroform, and water (2:2:1), at 25,000 rpm for 2 min using an IKA T18 Ultra-Turrax disperser (IKA-Werke GmbH, Staufen, Germany). Afterwards, the samples were centrifuged (5000× *g* for 10 min) to allow phase separation. The organic phase (chloroform) was then collected and transferred into new glass tubes. Next, a volume of 0.7 mL of lipid extract was poured into pre-weighed tubes and moved to a dry bath (60 °C) for overnight evaporation. After evaporation, the tubes were weighed and total lipid content was gravimetrically determined.

##### Ash Content

The ash content was determined by burning 50–60 mg of dried microalgal biomass in a muffle furnace (Sel horn R9-L, J.P. Selecta, Spain) at 525 °C for 8 h. The content was calculated according to Barreira et al. [35].

##### Carbohydrate Content

Carbohydrate content was obtained from mass balance after subtracting the protein, lipid, and ash contents. 

#### 2.3.2. Pigments’ Analyses

The extraction and analysis of pigments were performed according to Schüler et al. [40]. About 5–10 mg of dried microalgal biomass was resuspended in 1 mL of methanol, and about 0.6 mL of glass beads (425–600 µm) were added. The cells were disrupted in a Retsch MM 400 mixer mill (30 Hz for 3 min) followed by centrifugation (21,000× *g*, 3 min) to collect the supernatant. After transferring the supernatant to a new tube, the previous process was repeated until the pellet and the supernatant were colorless. Then, the extracts were combined, dried under a gentle nitrogen flow, resuspended in a known volume of MeOH, and filtered (0.22 µm). Total chlorophyll contents were determined spectrophotometrically using the sum of Equations (1) and (2) for *T. chui* [41] and Equation (3) for *N. oceanica* [42]: Chl *a* = 15.65 (A_666_–A_750_) − 7.34 (A_653_–A_750_)(1)
Chl *b* = 27.05 (A_653_–A_750_) − 11.21 (A_666_–A_750_)(2)
Chl *a* = 13.9 (A_665_–A_750_)(3)

Carotenoids were analyzed and quantified by a Dionex 580 HPLC System (DIONEX Corporation, Sunnyvale, CA, USA) consisting of a PDA 100 Photodiode-array detector, STH 585 column oven set to 20 °C, and a LiChroCART RP-18 (5 µm, 250 × 4 mm, LiChrospher) column. To this end, calibration curves of neoxanthin, violaxanthin, lutein, zeaxanthin, and *β*-carotene standards (Sigma-Aldrich, Lisbon, Portugal) were performed. The carotenoids were separated by a solvent gradient composed of solvent A, acetonitrile:water (9:1), and solvent B, ethyl acetate. The gradient program used was as follows: for 0–16 min increasing from 0–60% of solvent B, 16–30 min hold 60% of solvent B, 30–32 min increase to 100% of solvent B, and 32–35 min to 100% of solvent A. The injection volume of the extract and the standard was 100 µL. All carotenoids were detected at 450 nm and analyzed using the Chromeleon Chromatography Data System software (Version 6.3, ThermoFisher Scientific, Waltham, MA, USA).

#### 2.3.3. Fatty Acid Profile

The fatty acid profile was determined according to Lepage et al. [43] and further developed by Pereira et al. [44]. Briefly, 20–40 mg of freeze-dried microalgal biomass was weighed into derivatization vessels and resuspended in 1.5 mL of a solution of methanol:acetyl chloride (20:1, *v*/*v*). The samples were disrupted and homogenized with an IKA Ultra-Turrax disperser (2 min at 25,000 rpm), and 1 mL of *n*-hexane was added afterwards. All samples were heat treated in a water bath at 70 °C for 60 min. After cooling, they were transferred into centrifuge tubes, and 4 mL of *n*-hexane and 1 mL of distilled water were added. After vortexing the samples, they were centrifuged (2000× *g* for 5 min) for phase separation. The hexane fraction was transferred into new tubes and the process was repeated once again. Anhydrous sodium sulphate was used in excess to remove residual water. The samples were filtered through 0.22 µm PFTE filters and the solvent was evaporated under a gentle nitrogen flow until dry. Finally, all samples were resuspended in 0.5 mL of chromatography-grade hexane. 

Fatty acid methyl esters (FAME) were analyzed by a Bruker gas chromatographer coupled to a mass spectrometry system (Bruker SCION 456-GC, SCION TQ MS) equipped with a ZB-5MS capillary column (30 × 0.25 mm of internal diameter with 0.25 μm film thickness; Phenomenex), with helium as carrier gas (1 mL/min). The temperature program was set to 1 min at 60 °C, 30 °C/min to 120 °C, 4 °C/min to 250 °C, and 20 °C/min to 300 °C, held for 4 min, with an injection temperature of 300 °C in splitless mode. Supelco^®^ 37 component FAME Mix (Sigma-Aldrich, Sintra, Portugal) was used as a standard to prepare different calibration curves for FAME identification. The results are expressed as a percentage of total fatty acid (% TFA) contents. 

#### 2.3.4. Mineral Element Content

Mineral elements were determined by inductively coupled plasma optic emission spectrometry (ICP-OES iCap 7200 Duo, Thermo Fisher Scientific, Dreieich, Germany). The method was modified according to BVL method F 0042 [45], adopted from DIN EN 15510:2007 [46]. After incinerating 5 g of microalgal biomass at 550 °C (for P, Na, Mg, Ca, Fe, Cu, Mn) or 450 °C (for K), ash residues were dissolved in 10 mL nitric acid (20%; 65% Nitric acid, Cat. No. 1.00456.2500, Merck KGaA, Darmstadt, Germany) under slight boiling. After cooling, the solution was transferred into a 100 mL volumetric flask and topped up to the mark with demineralized water. P, Na, K, Mg, and Ca contents were determined by external calibration (phosphate standard solution, 1000 mg/L, Cat. No. 1.19898.0500; potassium standard solution, 1000 mg/L, Cat. No. 1.70230.0500; sodium standard solution, 1000 mg/L, Cat. No. 1.70238.0500; standard magnesium solution, 1000 mg/L, Cat. No. 1.70331.0100; calcium standard solution, 1000 mg/L, Cat. No. 1.19778.0500). Standard addition was performed to determine Fe, Cu, and Mn (iron standard solution, 1000 mg/L, Cat. No. 1.19781.0100; copper standard solution, 1000 mg/L, Cat. No. 1.19786.0100; manganese standard solution, 1000 mg/L, Cat. No. 1.19789.0100; Merck KGaA, Darmstadt, Germany). The measurement was performed with the following wavelengths: P, 213.618 nm; Na, 589.590 nm; Mg, 279.079 nm, Ca; 318.128 nm; Fe, 259.940 nm; Cu, 324.754 nm; Mn, 257.610 nm; and K, 766.490 nm.

#### 2.3.5. Microbial Safety Analyses

Microbial safety analyses were performed to investigate the total counts and the presence of (xerophilic) yeasts, molds, and *Escherichia coli* contaminations. Microalgal biomass (10 g) was suspended in 90 g of Maximum Recovery Diluent (MRD, Oxoid, Basingstoke, UK), followed by homogenization. The suspensions were diluted several times (10^−1^–10^−4^), and agar plates were inoculated with 100 µL of each dilution using the spread plate method. Only plates with 10 to 200 colonies were counted. Total counts were performed on plate count agar (PCA, Oxoid, Basingstoke, UK), and plates were incubated at 30 °C for 5 days following ISO 4833 [47]. Yeasts and molds were determined by incubating Oxytetracycline-Glucose Yeast Extract (OGYE, Oxoid, Basingstoke, UK) agar plates at 25 °C for 5 days, following the ISO 7954 guidelines [48]. Xerophilic yeasts and molds were determined by incubating Dichloran-glycerol 18% (DG18, Oxoid, Basingstoke, UK) agar plates at 25 °C for 5 days according to ISO 21527-2 [49]. The presence of *E. coli* was determined using RAPID’ *E. coli* 2 agar (Bio-Rad, Hercules, CA, USA), incubated at 44 °C for 24 h [50].

#### 2.3.6. Functional Properties

##### Water- and Oil-Holding Capacities

Water- and oil-holding capacities were determined according to Stone et al. [51] with minor modifications. Briefly, 0.5 g of microalgal powder was suspended in 10 mL of distilled water or sunflower oil in a 50 mL Falcon tube. During a time period of 30 min, samples were vortexed for 10 s every 5 min, followed by centrifugation at 1000× *g* for 15 min at room temperature (Sorvall LYNX 6000, Thermo Fisher Scientific, Waltham, MA, USA). After centrifugation, the supernatant was carefully decanted to remove any unbound water or oil, and the remaining pellet was weighed. The results were expressed as g of water or oil per g of powder (Equation (4)).
(4)$$WHC\;or\;OHC=\frac{w-w_{0}}{w_{0}}$$
where *w* is the weight of the pellet and *w*_0_ is the weight of microalgae powder. 

##### Foaming Properties

Foaming properties were determined according to Xiong et al. [52] with minor modifications. Briefly, 0.15 g of microalgal powder was suspended in 15 mL of distilled water. Samples were stirred magnetically for 30 min and whipped with a rotor-stator disperser Ultra-Turrax T25 (Ika, Staufen, Germany) at 8000 rpm for 1 min. Immediately after whipping, the foam was transferred to a 25 mL graduated cylinder. Foaming capacity (FC) was defined by the ratio of foam volume after 2 min and the initial sample volume (Equation (5)), while foaming stability (FS) was determined by the ratio of the foam volume after 10, 20, 40, or 60 min and the initial foam volume (Equation (6)).
(5)$$\mathrm{FC}\;\left[\%\right]=\frac{V_{0}}{15}\times 100$$
(6)$$\mathrm{FS}\;\left[\%\right]=\frac{V_{t}}{V_{0}}\times 100$$
where *V*_0_ is the foam volume after 2 min and *V_t_* is the foam volume at 10, 20, 40. or 60 min. 

##### Water Solubility Index

Microalgae cell solubility was determined according to Calderón-Castro et al. [53] with modifications. Briefly, 0.5 g of dried microalgae powder was suspended in 10 mL of distilled water in a 50 mL Falcon tube. Samples were magnetically stirred for 30 min, followed by centrifugation at 1800× *g* for 15 min at room temperature (Sorvall LYNX 6000, Thermo Fisher Scientific, Waltham, MA, USA). After centrifugation, the supernatant was decanted in a metal dish and the residue was weighed after drying for 12 h at 105 °C.
(7)$$WSI\;\left[\%\right]=\frac{w_{S}}{w_{0}}\times 100$$
where *w_S_* is the weight of the dried supernatant and *w*_0_ is the dry weight of the sample. 

##### Emulsion Capacity

Emulsifying capacity (EC) was determined according to Benelhadj et al. [54] with modifications. Suspensions containing 0.5%, 1%, and 2% (*w*/*v*) microalgae in distilled water were magnetically stirred for 30 min. For each concentration, 30 mL of suspension was added to 30 mL of sunflower oil (obtained from a local supermarket) and homogenized at 15,000 rpm with a rotor-stator disperser Ultra-Turrax T25 (Ika, Staufen, Germany) to form an oil-in-water emulsion. After homogenization, 10 mL of the emulsion was transferred into a 15 mL Falcon tube and immediately centrifuged at 1500× *g* for 5 min (Sorvall LYNX 6000, Thermo Fisher Scientific, Waltham, MA, USA). After centrifugation, the volume of the emulsified fraction was recorded and EC was expressed as a percentage of the emulsified volume.
(8)$$\mathrm{EC}\;\left[\%\right]=\frac{V_{E}}{V_{T}}\times 100$$
where *V_E_* is the volume of the emulsified fraction and *V_T_* is the total volume. 

### 2.4. Statistical Analyses

ANOVA with Tukey’s post hoc test was used to detect significant effects between the drying methods (FDc and SD I, II, and III) on response variables (proximate composition, pigments, fatty acids, minerals, microbial safety analyses, and functional properties) for each microalgal species. A significance level (α) of 0.05 was used for all performed tests. Statistical categories are represented with letters in graphs and tables. Data points are reported as means ± standard deviations. 

## 3. Results

### 3.1. Drying Process

#### 3.1.1. Solar-Drying Conditions

Solar-drying trials were performed within weeks 12 and 13 of 2021, where average outdoor temperatures ranged between 15.6 and 18.2 °C. No major environmental differences were observed except for solar radiation between the trials (*R_s_*; Table 1). R_s_ measurements revealed 58.66 and 45.63% lower average values during SD II compared to SD I and SD III, respectively. The average outdoor humidity ranged between 65.8 and 76.6% during the trials, while the average indoor humidity ranged from 28.4 to 32.2%. Automatic measurements of humidity and temperature are illustrated in Figure 2. Moreover, Figure 2 shows activities of the heater and the dehumidifier inside the hybrid solar dryer, correlated with average outdoor temperature and humidity. The customized algorithm maintained indoor temperatures between 23.9 and 49.3 °C during all trials (SD I, II, and III).

#### 3.1.2. Moisture Analyses

Regarding relative moisture loss during solar drying, significant differences were found between *T. chui* and *N. oceanica* biomass (Figure 3). Despite higher initial moisture contents of frozen *T. chui* biomass (78.22 ± 0.01%), *N. oceanica* (72.09 ± 0.01%) showed a significantly higher final moisture content, ranging between 7.02 ± 0.49 and 10.54 ± 0.15%, compared to that of *T. chui* (6.34 ± 0.07–7.46 ± 0.07%; Figure 3). On the other hand, moisture contents of freeze-dried samples were 3.51 ± 0.41% for *T. chui* and 0.66 ± 0.03 for *N. oceanica*.

### 3.2. Biomass Analyses

#### 3.2.1. Proximate Composition

Proteins, lipids, and carbohydrates revealed no significant differences between the different drying technologies for both species (Table 2). However, *N. oceanica* FDc showed a significant lower ash content than all solar-dried biomass.

#### 3.2.2. Pigment Analyses

The carotenoids neoxanthin, violaxanthin, lutein, and *β*-carotene as well as total chlorophyll contents of *T. chui* biomass were analyzed (Figure 4a). For *N. oceanica* biomass, the pigment analyses comprised violaxanthin, zeaxanthin, *β*-carotene, and total chlorophylls (Figure 4b). Freeze-dried *T. chui* biomass revealed the highest contents of the pigments neoxanthin (0.61 ± 0.02 mg/g) and violaxanthin (0.41 ± 0.01 mg/g), which were 14.33–19.09% higher (*p* < 0.05) compared with solar-dried *T. chui* biomass (SD I, II, and III; *p* < 0.05). On the other hand, lutein and *β*-carotene showed no statistically significant differences. However, when comparing total chlorophylls, significantly higher contents were found for FDc (23.24 ± 0.28 mg/g) than for SD I, II, and III (average: 19.06 ± 0.75 mg/g). 

Similar to *T. chui*, species-typical pigments *of N. oceanica* (Figure 4b) were affected by the drying methods. The contents of the major pigments violaxanthin and zeaxanthin were significantly higher in the FDc (0.29 ± 0.01 and 0.15 ± 0.00 mg/g, respectively), when compared with all SD samples (up to 0.20 ± 0.01 and 0.12 ± 0.01 mg/g, respectively). The contents of *β*-carotene of SD and FDc were not significantly different. In contrast, the total chlorophyll contents for all SD samples (ranging from 12.28 ± 0.16–14.33 ± 0.42 mg/g) were significantly lower than those of the FDc (20.56 ± 0.53 mg/g).

#### 3.2.3. Fatty Acid Profile

The fatty acid profile of *T. chui* was mainly composed of palmitic (C16:0), palmitidonic (C16:4*n*-3), oleic (C18:1*n*-9), and eicosapentaenoic (C20:5*n*-3, EPA) acids, while *N. oceanica* presented mainly eicosapentaenoic, palmitoleic (C16:1*n*-9), and palmitic (C16:0) acids. At concentrations below 10% of TFA, *T. chui* showed stearidonic (C18:4*n*-3), linoleic (C18:2*n*-6), γ-linolenic (C18:3*n*-6), palmitoleic, arachidonic (C20:4*n*-6), palmitolinolenic (C16:3*n*-3), heneicosylic (C20:1), and myristic (C14:0) acids, while *N. oceanica* presented arachidonic, myristic, linoleic, and oleic acids. However, there were no significant differences detected between the fatty acid profile of solar- (SD I, II, and III) and freeze-dried (FDc) *T. chui* and *N. oceanica* biomass (Table 3). 

#### 3.2.4. Mineral Contents

The analyses of minerals (total P, Na, K, Mg, Ca, total Fe, Cu, and Mn) revealed statistically significant differences between the two tested drying methodologies for some elements (Table 4). Potassium contents of freeze-dried *T. chui* (18.54 ± 0.03 mg/g) differed significantly from all solar-dried samples (17.28–17.73 mg/g), although the relative differences were only 4.6–7.3%. Freeze-dried *N. oceanica* (5.42 ± 0.03 mg/g) biomass showed minor differences in comparison with potassium contents of all solar-dried samples (4.92–5.09 mg/g). Moreover, magnesium quantities in freeze-dried *T. chui* biomass (11.46 mg/g) exceeded those of the solar-drying trials (10.91–11.04 mg/g) significantly. The most abundant mineral in *T. chui* and *N. oceanica* biomass was sodium, with 56.67–68.99 mg/g and 27.14–37.58 mg/g, respectively.

#### 3.2.5. Microbial Safety Analyses

Significant differences in microbial safety parameters between solar-dried samples (SD I, II, and III) and the freeze-dried control group (FDc) were detected. For *T. chui* biomass, total counts (PCA) were higher in all SDs (3.73 and 5.70 × 10^2^ CFU) compared to freeze-dried samples (1.10 × 10^2^ ± 1.00 × 10). Low numbers of (xerophilic) molds (<10^2^ CFU/g, no yeasts) were detected in both *T. chui* and *N. oceanica* SD samples. In the freeze-dried control sample, yeasts and molds were not detected above the limit of quantification (10 CFU/g). The total counts (PCA) for *N. oceanica* indicated significant differences between solar-drying trials (SD I, II, and III) and the FDc (Figure 5d). Total counts (PCA) for all treatments ranged between 5.57 × 10^3^ ± 5.13 × 10^2^ (FDc) and 2.40 × 10^5^ ± 1.00 × 10^4^. Molds detected in SD I and SD II did not present significant differences (*p* < 0.05, Figure 5f). The presence of *E. coli* was determined and revealed no detection above the limit of detection (10 CFU/g) for all solar- and freeze-dried samples.

#### 3.2.6. Functional Properties

Analysis of variances (ANOVA) fitted to functional properties (Figure 6) of solar- (SD I, II, and III) and freeze-dried (FDc) biomass revealed minor differences when comparing solar with freeze drying, except for the foaming capacity of *T. chui* samples. Concerning water-holding capacities (WHC), the FDc of *T. chui* (2.44 ± 0.06 g/g) was significantly higher than SD (between 2.05 and 2.10 g/g), while, for *N. oceanica* biomass, the opposite trend was observed, namely, SD samples represented higher values than the FDc. However, for oil-holding capacities (OHC), significant differences were only found for the FDc *N. oceanica* samples (1.26 ± 0.02 g/g) as compared with SD II (1.42 ± 0.06 g/g) and III (1.40 ± 0.03 g/g). The water-solubility index (WSI) showed no significant differences between each algae sample. Regarding foaming capacity, freeze-dried *T. chui* biomass showed significantly higher values (34.55 ± 3.33%) than SD I (18.98 ± 3.44%), SD II (18.98 ± 0.00%), and SD III (28.77 ± 0.00%). No foaming-capacity values were detected for solar- and freeze-dried *N. oceanica* biomass. Foaming stability decreased with time, and only non-significant differences were detected between the samples. For *N. oceanica*, no foaming-stability values were detected. Regarding emulsifying capacity, only the FDc of *T. chui* showed detectable values (10.36 ± 0.00%) at suspensions with 0.5% (g/100 mL) microalgal biomass. Emulsifying capacities for all SD and FDc samples of *T. chui* and *N. oceanica* increased gradually from suspensions with 1% to 5%. The FDc of *T. chui* showed significantly higher values for suspensions with 1 and 2% (28.50 ± 2.59% and 49.23 ± 0.00%, respectively) than all SD aliquots, ranging between 14.24–21.58% and 37.37–43.16%, respectively. Statistical analyses for suspensions with 5% *T. chui* biomass showed significantly lower values for the FDc group (51.82 ± 0.00%) than SD. Emulsifying capacities for 0.5% and 1% suspension of *N. oceanica* were not detected. Regarding suspensions with 2%, the FDc of *N. oceanica* showed significantly higher values (30.20 ± 0.00%) than SD I (10.75 ± 0.00%) and SD II (10.96 ± 0.00%). Moreover, minor differences between 5% suspensions of FDc and SD I and II (49.31–50.33%) compared with SD III (13.97 ± 2.79%) were found.

## 4. Discussion

Solar dryers of different types are gaining considerable attention for being economical and sustainable options for drying microalgal biomass [8,30,55,56]. Hence, this study investigated the effect of an indirect and hybrid solar dryer on biomass quality compared to a conventional freeze dryer (control), known to be one of the gentlest drying technologies for microalgae, preserving micro- and macronutrients [26,57]. Solar drying reduced moisture contents of *Tetraselmis chui* (initially 78.22 ± 0.01%) and *Nannochloropsis oceanica* (initially 72.09 ± 0.01%) down to 7.04 ± 0.61 and 8.76 ± 1.76%, respectively. This moisture reduction required a maximum residence time of 28 h, while freeze drying required 2-fold longer residence times (60 h). On the other hand, freeze-dried *T. chui* biomass showed significantly lower moisture contents (3.51 ± 0.41%), whereas *N. oceanica* biomass averaged 0.66 ± 0.33%. Since the drying process of frozen microalgal paste is controlled by the diffusion rate of internal moisture to the surface layer [58,59], species-dependent variations in biochemical profiles and rheological properties most likely influenced the drying behavior of the tested biomass. Moreover, case hardening of the biomass, which prevents core moisture from evaporating, might have also influenced the drying rates [57,59]. These phenomena could also explain the significantly higher final moisture contents of solar-dried *N. oceanica* than *T. chui* and of all solar-dried samples compared to freeze-dried.

No significant differences between the applied drying technologies were detected when comparing protein, lipid, and carbohydrate contents of *T. chui* and *N. oceanica* biomass. These observations vary from Stramarkou et al. [60], who found higher protein contents in freeze-dried *Chlorella vulgaris* biomass, or Desmorieux and Hernandez [61], who found lower protein contents in freeze-dried *Spirulina* sp. biomass as compared to solar-dried biomass. Nevertheless, our results highlight the potential of the indirect and hybrid solar-drying technology as a preferred option to dry biomass for general uses, such as human nutrition.

Conversely, significant differences were found for pigment contents when comparing solar- and freeze-dried biomass. Violaxanthin, neoxanthin, and zeaxanthin contents in solar-dried biomass were significantly lower than freeze-dried biomass, which might be an effect of heat-induced degradation of these xanthophylls [62]. Certain pigments are generally sensitive to heat, oxygen, and light, leading to degradation [63]. These differences were not found for lutein and *β*-carotene, which could be related to the higher heat resistance of these carotenoids and the higher concentrations present in the biomass [64]. Compared to the freeze-dried control group, higher lutein amounts were detected for SD I and SD II, whereas all solar-dried samples showed higher *β*-carotene contents. This suggests that dehydration in an indirect, hybrid solar dryer may improve the carotenoid extractability of microalgal biomass [64]. However, significantly lower total chlorophyll contents were recovered from solar-dried *T. chui* and *N. oceanica* biomass compared with the freeze-dried control group, which is in accordance with the studies done by Shekarabi et al. [65]. Chlorophyll degradation is probably a result of oxidative degradation and has previously been found in other studies that evaluated different drying technologies for *Chlorella vulgaris*, *Chlorella* (*Auxenochlorella*) *pyrenoidosa*, and *Nannochloropsis* (*Microchloropsis*) *salina* [60,64].

Regarding the fatty acid profile, the solar-dried biomass showed no significant differences for saturated, monounsaturated, and polyunsaturated (PUFAs) fatty acids compared to the freeze-dried control. In particular, no significant differences were found for the omega-3 fatty acid EPA, suggesting that no oxidation of this PUFA occurred during the solar-drying process. Therefore, the indirect and hybrid solar-drying technology appears promising for preserving high-value fats. 

Except for sodium contents, only minor differences among the mineral profiles of solar- (SD I, II, and III) and freeze-dried *T. chui* and *N. oceanica* biomass were detected (<10% among all samples). This may be explained by the high-temperature stabilities of the analyzed macro- and micronutrients. These minor differences may originate from the high accuracy of the ICP-OES method used or the natural variance across the different dryings (significant differences between trials). Sodium was the most abundant mineral in *T. chui* and *N. oceanica* biomass, with 56.67–68.99 mg/g and 27.14–37.58 mg/g, respectively. According to the European Regulation (EC) No. 1169/2011 [66], this results in salt contents of approx. 14.17–17.25 g/100 g for *T. chui* and 6.78–9.40 g/100 g for *N. oceanica*. These amounts are relatively high compared to other foods, e.g., cheddar cheese (6.54 mg/g sodium, 1.64 g/100 g salt) or tortilla chips (4.29 mg/g sodium, 1.1 g/100 g salt) [67]. Adults’ daily salt reference intake is 6.00 g, limiting marine microalgae consumption. Nevertheless, biomass can be pre-treated to remove the excess salt (diafiltration, for instance). The significant differences of up to 27.80% in sodium contents between solar- and freeze-dried biomass are most likely related to the experimental drying setups and the used materials. While plain stainless-steel trays were used for freeze drying, elevated grids (on top of stainless-steel trays) were used for solar drying. Since exudate can run off through the grids, sodium ions in the water might have separated from the remaining biomass by gravity. However, the results suggest that hybrid solar-drying technologies can be applied to preserve mineral element contents along the drying process of microalgal biomasses.

Since the solar-drying device uptakes unfiltered ambient air through the intake ventilation, the risk of external contamination was given. Compared to freeze drying, total counts of microbial activities were significantly higher for the solar-dried biomass of both microalgae species. Moreover, no (xerophilic) molds were detected in the freeze-dried samples, contrary to some solar-dried samples, albeit in low numbers (<10^2^ CFU/g). These differences in microbial load might be attributed to higher temperatures during the solar-drying process (23.91–49.30 °C), which are optimal conditions for mesophilic microbial growth compared to sub-zero temperatures during freeze drying [68]. Varying weather conditions during the solar-drying experiment could also explain the significant differences in total counts between the solar-dried samples (SD I, II, and III) for *N. oceanica*. Although the solar-dried biomass generally revealed higher microbial activity, no *E. coli* was detected in any sample, which is usually a primary indicator for (fecal) food contaminations [69]. According to Uyttendaele et al. [70], dried plant products (Category 4F) are considered safe if xerophilic mold counts do not exceed 3 × 10^5^ CFU/g (target: 3 × 10^4^ CFU/g) and *E. coli* counts are below 3 × 10^3^ CFU/g (target: 3 × 10^2^ CFU/g). Therefore, this study’s solar- and freeze-dried microalgal biomass can be considered food safe because they meet the aforementioned target guidelines.

Regarding techno-functional properties, *T. chui* and *N. oceanica* biomass showed significant differences in their water- and oil-holding capacities, which can be explained by slight differences in their composition (protein and water contents) [71]. The foaming and emulsifying capacities of freeze-dried *T. chui* samples were significantly higher than those of SD samples. In addition, *N. oceanica* biomass did not show any foaming and emulsifying properties at low concentrations. The stabilization of foams and emulsions is achieved by proteins and long-chain carbohydrates interacting with hydrophobic and hydrophilic groups, which causes a reduction in the interfacial tension between air and water or water and oil, respectively [72,73,74]. In addition to the adsorption of a sufficient concentration on the interface, which depends on the viscosity of the system and the solubility of the molecules, the emulsification technique, the surface hydrophobicity, and the molecular flexibility also have an impact on the formation of stable interfacial films [75,76]. Furthermore, the density and distribution of charge at the interface influence the electrostatic repulsion, which must be in balance with attractive forces, such as van der Waals forces, for the emulsion or foam to remain stable [77]. From this, it can be deduced that *N. oceanica* may (1) be less adsorbed at the interphase than *T. chui* due to the lower water solubility index, (2) have lower long-chain carbohydrate or protein contents on its cell wall, (3) have an improper surface charge, and/or (4) be too inflexible or unable to form several bonds along the interface due to its small size. Additionally, Guil-Guerrero et al. [78] found significantly lower emulsification capacities for defatted *Nannochloropsis* spp. compared to soybean flower, *Porphyridium cruentum*, and *Phaeodactylum tricornutum*. Interestingly, differences in foaming capacity and emulsion capacity resulting from solar drying cannot be related to the higher drying temperature compared to freeze drying because higher temperatures are known to yield better foaming and emulsifying properties as a result of protein unfolding exposing hydrophobic sites, which improves interfacial activity [79,80]. Slight differences between the FDc and SD samples again may be explained by compositional differences and changes on the molecular surface. Further analyses, e.g., determination of molecular hydrophobicity, surface charge, and interfacial tension, are needed to clarify if conformational changes in the cell walls were caused by the different drying techniques. Overall, solar drying can be considered a promising technology to provide high-quality biomass for food and feed markets.

## 5. Conclusions

The present study showed the promising application of an indirect and hybrid solar dryer to dehydrate microalgal paste. Since freeze drying is a reliable technology in the microalgae industry, multiple quality parameters of solar-dried *Tetraselmis chui* and *Nannochloropsis oceanica* biomass were compared with freeze-dried samples (control). We investigated the effect of both drying methods and observed that total proteins, carbohydrates, fats, and fatty acid profiles showed no significant differences. Conversely, freeze-dried samples showed significantly higher contents of certain pigments (e.g., total chlorophylls), while deviations of mineral profiles remained minor. Microbial safety analyses indicated that solar-dried microalgal biomass could be considered food grade. In comparison with freeze-dried biomass, solar-dried biomass showed significant differences in water- and oil-holding capacities, as well as in foaming and emulsifying capacities. Moreover, the foaming stability of *T. chui* biomass differed significantly between the SD and FDc. In summary, the obtained results appear promising for using solar-dried *T. chui* and *N. oceanica* biomass for innovative and sustainable food and feed products. Further research should address the energy efficiency of the solar dryer since only approx. 1.36% of the maximum capacity of the solar dryer was used in this study. Nevertheless, our study demonstrates that indirect hybrid solar drying holds a high potential to be a viable alternative to conventional drying technologies, such as freeze or spray drying.

## Figures and Tables

**Figure 1 foods-11-01873-f001:**
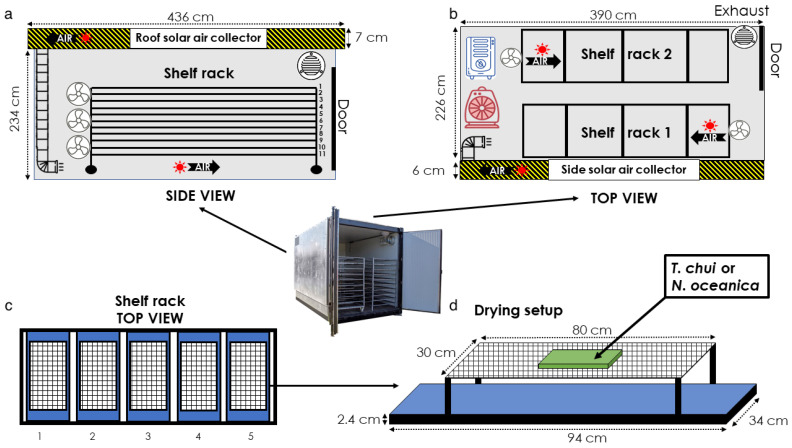
Technical diagram of the solar dryer and the experimental setup: solar drier side view (**a**), solar drier top view (**b**), top view of the shelf rack (**c**), and drying setup scheme (**d**).

**Figure 2 foods-11-01873-f002:**
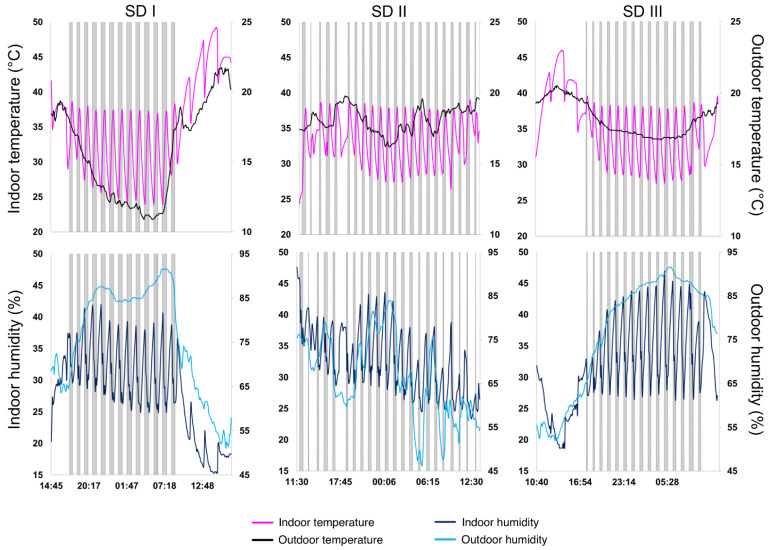
Plotted activity log (SD I, II, and III) of the hybrid solar dryer, including outdoor temperature and humidity obtained from the weather station. The time intervals when the heater (upper panel) or the dehumidifier (lower panel) ran are shown as gray, vertical bars.

**Figure 3 foods-11-01873-f003:**
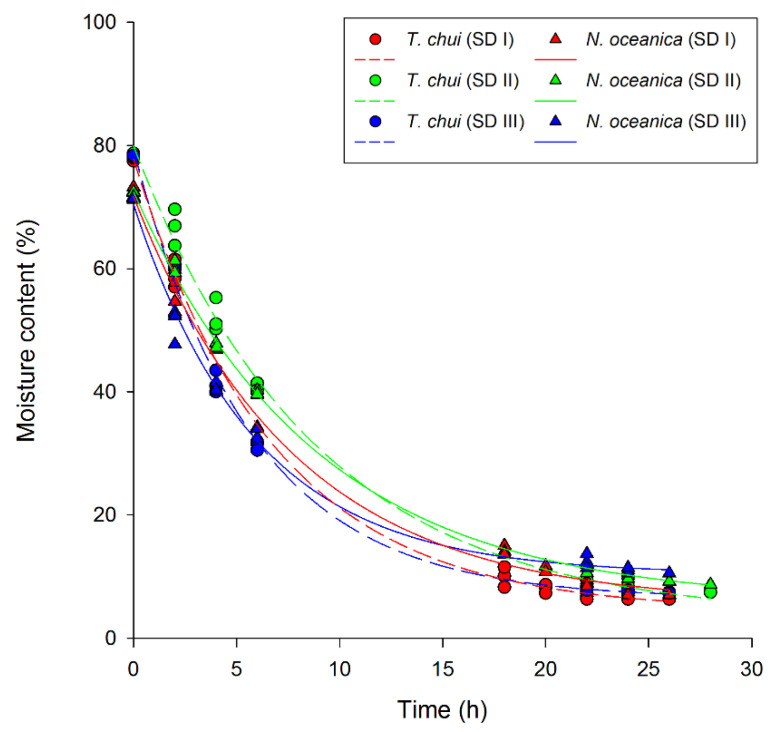
Relative moisture loss of solar-dried *Tetraselmis chui* (circles and dashed lines) and *Nannochloropsis oceanica* (triangles and solid lines) biomass during three independent drying trials (SD I, II, and III).

**Figure 4 foods-11-01873-f004:**
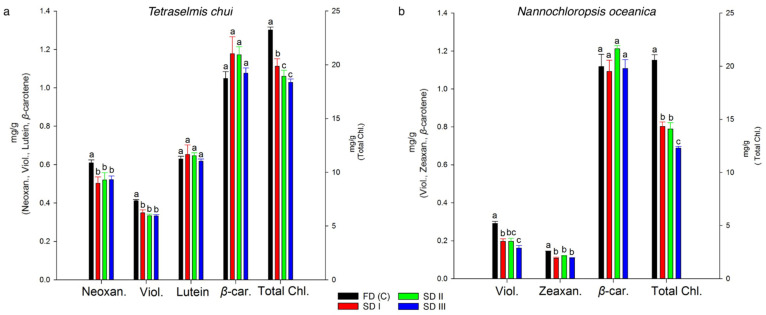
Total neoxanthin, violaxanthin, lutein, *β*-carotene, and total chlorophyll contents of solar- (SD I, II, and III) and freeze-dried (FDc) *Tetraselmis chui* (**a**). Total violaxanthin, zeaxanthin, *β*-carotene, and total chlorophyll contents of solar- (SD I, II, and III) and freeze-dried (FDc) *Nannochloropsis oceanica* (**b**). Different letters represent significant differences detected by Tukey’s post hoc range test (HSD, ANOVA). Data points for solar- and freeze-dried samples are shown as mean ± Std Dev (*n* = 3). Error bars represent standard deviations.

**Figure 5 foods-11-01873-f005:**
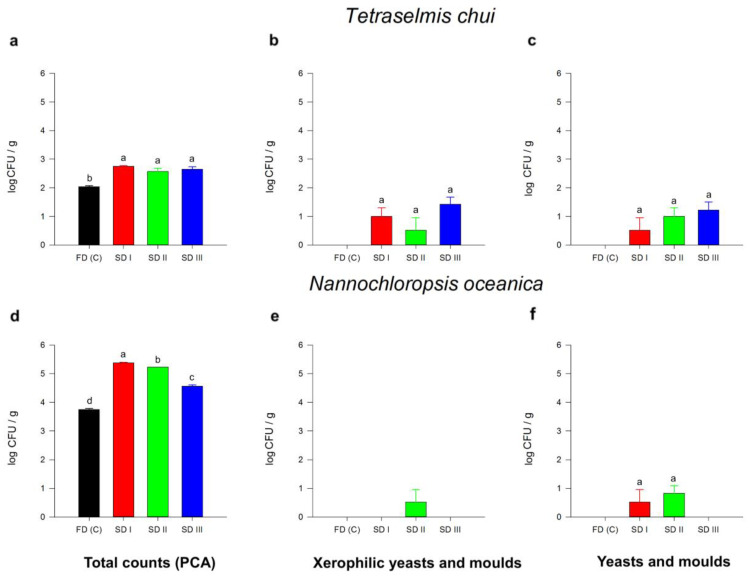
Total counts (PCA; **a**,**d**), xerophilic yeasts and molds (**b**,**e**), and yeasts and molds (**c**,**f**) of solar- (SD I, II, and III) and freeze-dried (FDc) *Tetraselmis chui* (**a**–**c**) and *Nannochloropsis oceanica* (**d**–**f**) biomass. Different letters represent significant differences detected by Tukey’s post hoc range test (HSD, ANOVA). Error bars represent standard deviation (Std Dev). Data points for solar- and freeze-dried samples are shown as mean ± Std Dev (*n* = 3).

**Figure 6 foods-11-01873-f006:**
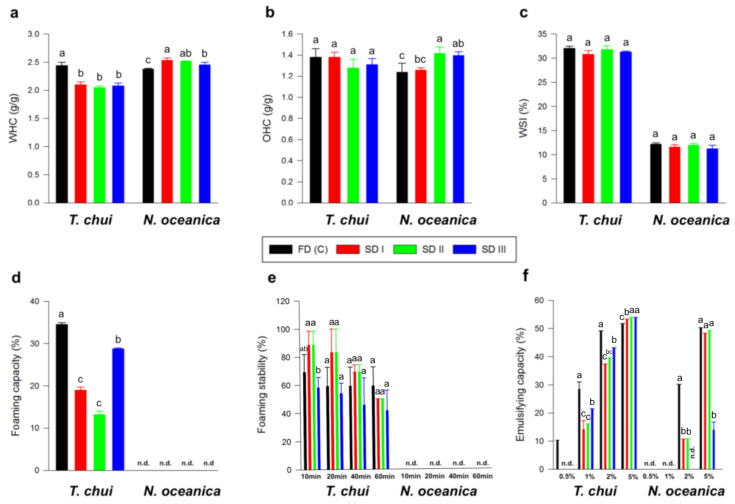
Water-holding capacity (WHC) (**a**), oil-holding capacity (OHC) (**b**), water solubility index (WSI) (**c**), foaming capacity (**d**), foaming stability (**e**), and emulsifying capacity (**f**) of solar- (SD I, II, and III) and freeze-dried (FDc) *Tetraselmis chui* and *Nannochloropsis oceanica* biomass. Different letters represent significant differences among samples from the same species detected by Tukey’s post hoc range test (HSD, ANOVA). Error bars represent standard deviation (Std Dev). Data points for solar- and freeze-dried samples are shown as mean ± Std Dev (*n* = 3). N.d., not detected.

**Table 1 foods-11-01873-t001:** Values for indoor and outdoor conditions.

Parameters	SD I	SD II	SD III
Indoor temperature min.–max. (°C)	23.9–49.3	24.3–39.0	27.4–46.0
Average indoor temperature (°C)	35.4	33.5	35.3
Indoor humidity min.–max. (%)	15.3–41.9	23.4–47.6	18.7–46.9
Average indoor humidity (%)	28.4	32.1	32.2
Outdoor temperature min.–max. (°C)	10.9–21.8	16.2–20.1	16.7–20.5
Average outdoor temperature (°C)	15.63	18.20	18.21
Outdoor humidity min.–max. (%)	51.1–91.7	46.1–84.0	51.9–91.6
Average outdoor humidity (%)	76.2	65.8	76.6
Average solar radiation (*R_s_*; W.m^−2^)	275.3	114.1	209.9

**Table 2 foods-11-01873-t002:** Proximate composition (proteins, total lipids, carbohydrates, and ashes) of solar- (SD I, II, and III) and freeze-dried (control, FDc) *Tetraselmis chui* and *Nannochloropsis oceanica* biomass. Different letters represent significant differences among samples from the same species detected by Tukey’s post hoc range test (HSD, ANOVA). Data points for all samples are shown as mean ± standard deviation (Std Dev) (*n* = 3).

Proximate Composition (%)	*Tetraselmis chui*				*Nannochloropsis oceanica*			
	SD I	SD II	SD III	FDc	SD I	SD II	SD III	FDc
Proteins	38.53 ± 0.78 ^a^	37.80 ± 0.64 ^a^	38.01 ± 0.88 ^a^	37.93 ± 0.11 ^a^	46.72 ± 0.31 ^a^	46.01 ± 0.94 ^a^	48.09 ± 0.41 ^a^	47.39 ± 2.18 ^a^
Total lipids	18.61 ± 1.46 ^a^	18.40 ± 1.62 ^a^	18.24 ± 0.50 ^a^	16.75 ± 1.62 ^a^	21.46 ± 2.78 ^a^	24.69 ± 1.95 ^a^	23.25 ± 1.97 ^a^	24.25 ± 3.37 ^a^
Carbohydrates	16.73 ± 1.27 ^a^	16.06 ± 1.05 ^a^	16.27 ± 1.18 ^a^	15.49 ± 1.78 ^a^	17.11 ± 2.54 ^a^	14.69 ± 2.61 ^a^	14.24 ± 1.62 ^a^	15.28 ± 4.25 ^a^
Ashes	28.46 ± 0.06 ^a^	30.44 ± 0.06 ^a^	30.24 ± 0.05 ^a^	30.86 ± 0.10 ^a^	14.71 ± 0.28 ^b^	14.62 ± 0.45 ^b^	14.42 ± 0.15 ^b^	13.08 ± 0.16 ^a^

**Table 3 foods-11-01873-t003:** Fatty acid profile of solar- (SD I, II, and III) and freeze-dried (FDc) *Tetraselmis chui* and *Nannochloropsis oceanica* biomass. No significant differences among samples from the same species were detected by Tukey’s post hoc range test (HSD, ANOVA). Data points for solar- and freeze-dried samples are shown as mean ± Std Dev (*n* = 3). N.d., not detected; SFA, saturated fatty acids; MUFA, monounsaturated fatty acids; PUFA, polyunsaturated fatty acids.

Fatty Acid (%)	*Tetraselmis chui*				*Nannochloropsis oceanica*			
	SD I	SD II	SD III	FDc	SD I	SD II	SD III	FDc
C14:0	1.12 ± 0.22	1.07 ± 0.15	1.05 ± 0.17	1.16 ± 0.18	4.85 ± 1.67	4.18 ± 0.74	4.20 ± 0.77	4.22 ± 0.61
C16:0	20.48 ± 1.88	20.58 ± 1.87	20.67 ± 1.68	22.41 ± 1.91	19.95 ± 5.60	18.09 ± 3.52	17.79 ± 3.29	18.11 ± 2.89
∑ SFA	21.59 ± 2.10	21.65 ± 2.03	21.72 ± 1.84	23.57 ± 2.09	24.80 ± 7.26	22.28 ± 4.26	21.99 ± 4.06	22.32 ± 3.50
C16:1*n*-9	3.96 ± 1.92	3.84 ± 1.20	3.02 ± 1.97	4.12 ± 1.22	25.30 ± 7.68	22.03 ± 3.33	21.80 ± 3.43	21.94 ± 2.85
C18:1*n-9*	17.29 ± 0.73	17.78 ± 0.49	17.96 ± 0.60	16.09 ± 4.42	3.61 ± 0.63	2.70 ± 0.80	2.95 ± 0.71	3.37±0.29
C20:1	1.14 ± 0.68	1.20 ± 0.70	1.42 ± 0.34	1.48 ± 0.33	n.d.	n.d.	n.d.	n.d.
∑ MUFA	22.39 ± 3.34	22.82 ± 2.40	22.40 ± 2.91	21.69 ± 5.97	28.9 ± 8.31	24.74 ± 4.13	24.75 ± 4.14	25.31 ± 3.14
C16:3*n*-3	1.56 ± 0.20	1.57 ± 0.16	1.58 ± 0.20	1.58 ± 0.17	n.d.	n.d.	n.d.	n.d.
C16:4*n*-3	19.32 ± 1.01	19.43 ± 0.52	19.87 ± 0.96	19.49 ± 1.10	n.d.	n.d.	n.d.	n.d.
C18:2*n*-6	5.09 ± 0.18	5.17 ± 0.11	5.24 ± 0.20	5.36 ± 0.40	4.61 ± 0.89	3.89 ± 0.34	3.92 ± 0.39	4.22 ± 0.21
C18:3*n*-6	5.08 ± 0.39	5.06 ± 0.29	5.17 ± 0.34	5.14 ± 0.58	n.d.	n.d.	n.d.	n.d.
C18:4*n*-3	9.03 ± 0.67	9.01 ± 0.38	9.19 ± 0.57	8.89 ± 0.92	n.d.	n.d.	n.d.	n.d.
C20:4*n*-6	1.84 ± 0.26	1.70 ± 0.23	1.62 ± 0.24	1.62 ± 0.29	10.31 ± 0.66	8.69 ± 1.24	9.20 ± 0.60	9.31 ± 0.34
C20:5*n*-3	12.12 ± 3.50	11.47 ± 3.15	11.15 ± 3.31	10.62 ± 3.58	28.72 ± 9.22	37.95 ± 6.70	37.29 ± 7.25	36.37 ± 7.27
∑ PUFA	54.04 ± 6.11	53.42 ± 4.85	53.82 ± 5.83	52.69 ± 7.05	43.64 ± 10.76	50.53 ± 8.28	50.41 ± 8.24	49.89 ± 7.82

**Table 4 foods-11-01873-t004:** Mineral contents of solar- (SD I, II, and III) and freeze-dried (FDc) *Tetraselmis chui* and *Nannochloropsis oceanica* biomass. Different letters represent significant differences among samples from the same species detected by Tukey’s post hoc range test (HSD, ANOVA). Data are shown as mean ± Std Dev (*n*=3).

Minerals (mg/g)	*Tetraselmis chui*				*Nannochloropsis oceanica*			
	SD I	SD II	SD III	FDc	SD I	SD II	SD III	FDc
Phosphorus	10.73 ± 0.06 ^b^	10.67 ± 0.02 ^ab^	10.75 ± 0.08 ^b^	10.53 ± 0.07 ^a^	11.44 ± 0.07 ^a^	11.51 ± 0.06 ^a^	11.48 ± 0.08 ^a^	11.26 ± 0.34 ^a^
Sodium	56.67 ± 2.28 ^d^	59.29 ± 1.57 ^c^	60.28 ± 0.04 ^b^	68.99 ± 0.45 ^a^	30.22 ± 0.44 ^b^	28.84 ± 0.16 ^a^	27.14 ± 0.80 ^a^	37.58 ± 0.24 ^a^
Potassium	17.28 ± 0.16 ^a^	17.73 ± 0.04 ^a^	17.64 ± 0.39 ^a^	18.54 ± 0.03 ^b^	5.07 ± 0.03 ^ab^	5.09 ± 0.09 ^b^	4.92 ± 0.06 ^a^	5.42 ± 0.03 ^c^
Magnesium	10.98 ± 0.07 ^a^	10.91 ± 0.12 ^a^	11.04 ± 0.10 ^a^	11.46 ± 0.07 ^b^	7.92 ± 0.08 ^a^	7.93 ± 0.07 ^a^	7.96 ± 0.12 ^a^	8.10 ± 0.03 ^a^
Calcium	16.46 ± 0.07 ^b^	16.37 ± 0.16 ^a^	16.02 ± 0.23 ^a^	16.33 ± 0.06 ^ab^	4.10 ± 0.03 ^b^	4.06 ± 0.03 ^ab^	4.05 ± 0.04 ^ab^	4.01 ± 0.04 ^a^
Iron	2.75 ± 1.55 ^b^	2.75 ± 0.31 ^b^	2.78 ± 0.78 ^b^	2.68 ± 1.92 ^a^	0.72 ± 2.71 ^a^	0.70 ± 0.91 ^a^	0.72 ± 0.06 ^a^	0.70 ± 0.48 ^a^
Copper	0.01 ± 0.00 ^ab^	0.01 ± 0.00 ^ab^	0.01 ± 0.00 ^b^	0.01 ± 0.00 ^a^	0.02 ± 0.00 ^a^	0.02 ± 0.00 ^a^	0.02 ± 0.00 ^b^	0.02 ± 0.00 ^a^
Manganese	0.11 ± 0.00 ^c^	0.10 ± 0.00 ^b^	0.11 ± 0.00 ^c^	0.10 ± 0.00 ^a^	0.04 ± 0.00 ^a^	0.04 ± 0.00 ^a^	0.05 ± 0.00 ^b^	0.04 ± 0.00 ^a^

## Data Availability

Data is contained within the article.

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
