# Peer review of "Drying Microalgae Using an Industrial Solar Dryer: A Biomass Quality Assessment"

_foods, 2022, doi:10.3390/foods11131873_

Round 1

Reviewer 1 Report

The submitted article is comparing the biochemical compositions and physical properties of solar and freeze-dried T. chui and N. oceanica biomass. Microbial food safety analyses of the dried biomass were performed to evaluate the suitability for food products. The article is well presented and the topic is relevant for the selected journal. The results are adequately interpreted and compared to the literature. The authors should correct the following:

Page 1, There are some technical issues that need to be addressed, provide mails of all authors, organize keywords, add citations,

Page 3,  line 111, please add the moisture content of the T.chui and N.oceanica biomass after  Freeze-drying.

Page 3, line 139, please add the moisture content of the T.chui and N.oceanica biomass after solar drying.

Add borders to Table 1-4, according to the instructions for authors.

Figure 5 should be centered to be visible.

Some minor English revisions are required.

Author Response

Comment: Page 1, There are some technical issues that need to be addressed, provide mails of all authors, organize keywords, add citations,

Answer: we appreciate the reviewers’ suggestions and we have revised the manuscript in order to clarify the points mentioned above (provide mails of all authors, organize keywords). Regarding the comment “add citation”, we were wondering in which section of the manuscript more citations are needed/or if there are any mistakes with the citations used for this study. Two citations were added to line 196. We would greatly appreciate it if the reviewer could inform us if more citations should be added or if any more changes should be done.

Comment: Page 3,  line 111, please add the moisture content of the T.chui and N.oceanica biomass after  Freeze-drying.

Answer: while we appreciate the reviewer’s feedback, in our point of view, the moisture contents of T.chui and N.oceanica biomass after freeze-drying should be mentioned only in the results section. Therefore, we added moisture contents of the T.chui and N.oceanica biomass after freeze-drying to page 9, lines 401 – 403 (section 3.1.2). We hope that these changes are sufficient and you agree with the new version of the manuscript.

Comment: Page 3, line 139, please add the moisture content of the T.chui and N.oceanica biomass after solar drying.

Answer: We appreciate the reviewer’s suggestion. Like in the above-mentioned answer, we consider final moisture contents as results of the study. Therefore, moisture contents of T.chui and N.oceanica biomass after solar-drying were already mentioned in section 3.1.2 of the manuscript (“Moisture analyses”).

Comment: Add borders to Table 1-4, according to the instructions for authors.

Answer: all tables have been adapted according to the journal guidelines.

Comment: Figure 5 should be centered to be visible.

Answer: figure 5 has been adapted according to the journal guidelines.

Comment: Some minor English revisions are required.

Answer: the new version of the manuscript was spell-checked once again and minor corrections were done.

Reviewer 2 Report

This study aimed to apply alternative drying technologies to lower the current cost of microalgal biomass with high quality and check its probable use by comparing with the current technologies.

Drying microalgae is an important challenge since the use of biomass without degradation for the production of valuable products for many industries is essential. In this respect, this study is novel and interesting. It proposes an alternative way of drying process and make a comparison with the present drying methods. They make it by comparing the biochemical and physical properties of the biomasses (T. chu and N. oceanica).

The literature review is current and well summarized.

The diagrams and Figures are very well for explaining the processes and the results.

The results were well presented and easy to understand.

The discussions were in accordance with the results and the literature supported.

Line 19: Please revise the sentence as “Microalgae are considered future proof bioresources of proteins, lipids, carbohydrates, pigments, and a variety of functional biomolecules for food and feed industries.

Line 178: Please insert the relevant references in addition to Reference 35. In addition it will be better to check the updated version of the second one.

1.     Uslu, L., DURMAZ, Y., Duyar, H., & Bandarra, N. (2013). Fatty Acids, alpha-Tocopherol and Proximate Composition of Four Red Macroalgae in the Sinop Bay (Turkey). Journal of Animal and Veterinary Advances, 12(1).

2.     AOAC, 1990. Official Methods of Analysis of the Association of Analytical. 15th Edn. AOAC, Washington, DC, USA.

Author Response

Comment: Line 19: Please revise the sentence as "Microalgae are considered future proof bioresources of proteins, lipids, carbohydrates, pigments, and a variety of functional biomolecules for food and feed industries.

Answer: Thank you for pointing this out. The abovementioned sentence was revised and changed to make it clearer for the reader. We appreciate the reviewer’s suggestions.

Comment: Line 178: Please insert the relevant references in addition to Reference 35. In addition it will be better to check the updated version of the second one.

  1. Uslu, L., DURMAZ, Y., Duyar, H., & Bandarra, N. (2013). Fatty Acids, alpha-Tocopherol and Proximate Composition of Four Red Macroalgae in the Sinop Bay (Turkey). Journal of Animal and Veterinary Advances, 12(1).
  2. AOAC, 1990. Official Methods of Analysis of the Association of Analytical. 15thEdn. AOAC, Washington, DC, USA.

Answer: We appreciate the reviewer's suggestions, and we have added the references mentioned above.